# Cognitive Loop of Thought: Reversible Hierarchical Markov Chain for Bidirectional Self-Verifying Reasoning

## Abstract

Multi-step Chain-of-Thought (CoT) has significantly enhanced the mathematical reasoning capabilities of large language models by leveraging clear reasoning steps and task-specific logical structures. However, with the widespread adoption of Long CoT, the number of reasoning steps often exceeds the system's manageable limits. To tackle these problems, existing approaches attempt to reduce redundancy in KV Cache by introducing Markov chain-like reasoning structures, thereby improving inference efficiency. Nonetheless, such Markov chain-based reasoning methods introduce two critical issues: Lack of memory and Limited backward reasoning capability. To address these limitations, we propose a novel Chain-of-Thought framework based on Reversible Hierarchical Markov Chain (RHMC), termed Cognitive Loop of Thought (CLoT), and a backward reasoning dataset CLoT-Instruct. In CLoT, the original problem is decomposed into sub-problems with hierarchical dependencies and modeled as a hierarchical Markov chain based on the number of dependencies. Humans typically revisit and verify their reasoning steps after reaching a conclusion to avoid errors. Inspired by this cognitive behavior, we introduce a similar backward verification mechanism at each layer. Moreover, when all higher-level (multi-dependency) sub-problems are verified as correct, we prune the remaining lower-level (fewer-dependency) sub-problems. CLoT effectively mitigates error propagation along the reasoning path and enhances the robustness of the entire reasoning process. We conduct experiments on four mathematical reasoning benchmarks, demonstrating the effectiveness of CLoT. Notably, on the AddSub dataset, when applied to the GPT-4o-mini model, CLoT achieves an accuracy of 99.0% , outperforming traditional CoT and CoT-SC by 4.1% and 2.9%. Our code is publicly available at: https://anonymous.4open.science/r/CLoT-7EBD.

## 1 Introduction

Large Language Models (LLMs) have demonstrated significant advancements across a broad spectrum of Natural Language Processing (NLP) tasks, such as question answering, automatic summarization, and machine translation Achiam et al. (2023); Chowdhery et al. (2023); Touvron et al. (2023); Huang et al. (2023); Zhao et al. (2023). Despite these achievements, LLMs continue to exhibit notable limitations in reasoning performance when compared to human cognitive capabilities. Empirical evidence suggests that merely increasing model scale through parameter expansion is insufficient to bridge the gap in complex reasoning abilities between current LLMs and human-level intelligence. Zhou et al. (2024).

To enhance the model's reasoning capabilities, researchers have explored various strategies to enhance the logical coherence and step-by-step inference processes of LLMs. One of the most influential approaches is Chain-of-Thought (CoT) reasoning Kojima et al. (2022), which enables models to generate intermediate reasoning steps before arriving at a final answer. By mimicking human-like problem-solving patterns, CoT not only improves model performance on complex reasoning tasks, but also enhances the interpretability of model decisions. The transparency afforded by explicit reasoning traces allows users to inspect, validate, and potentially correct the model's logic, thereby fostering trust and facilitating debugging.

However, when facing real-world application scenarios, CoT still encounters the challenge that intermediate reasoning steps may contain errors. To address this challenge, one approach to improve performance is intrinsic self-correction Kamoi et al. (2024); Pan et al. (2024), which allows the model to check and revise its own generated answers without external feedback—a process highly analogous to human thinking. Nevertheless, negative views on self-correction also exist. Stechly et al. (2023); Tyen et al. (2024); Jiang et al. (2024) finds that large language models are even unable to determine the correctness of answers, often changing correct answers into incorrect ones or failing to correct originally erroneous responses. The debates in prior research indicate that the self-correction mechanisms of large models still lack in-depth exploration.

To address this challenge and overcome the limitations of training large models for verification, we propose a novel reverse self-verification Chain-of-Thought method, named CLoT (Cognitive Loop of Thought), to validate the results of generated answers. In many decision-making tasks, humans often perform self-validation on inferred conclusions to reduce errors. In this paper, we aim to enable LLMs to learn a similar self-verification capability. Inspired by reversible Markov chains, we treat one forward refinement step and one backward verification step as a single loop, and iteratively execute this loop to ensure the accuracy of intermediate reasoning steps in the cot. Specifically, in the forward refinement phase, LLMs use CoT to progressively generate a series of sub-questions and their corresponding answers. Subsequently, each sub-question undergoes backward verification: the original conditions of the sub-question are treated as unknowns, while the generated answer is treated as a known condition, and reasoning is performed again in reverse. The backward inference result is then compared with the original condition. If the values match, the sub-question and its answer are considered correct, and the process proceeds to verify the next sub-question in sequence. If they do not match, the model is prompted to analyze the cause—determining whether the backward verification is inapplicable or whether the original answer was incorrect. By verifying all sub-questions in this manner, our method minimizes the likelihood of errors occurring in the intermediate reasoning process. We conduct experiments using gpt-4o-mini and gpt-4-1106-preview on multiple open-source datasets, involving mathematical reasoning, commonsense reasoning, and logical reasoning tasks, achieving results that surpass the baselines. Our contributions are summarized as follows:

- We propose a novel self-reflective verification method for reasoning chains, named CLoT (Cognitive Loop of Thought). CLoT is capable of verifying both intermediate steps and the final answer, as well as analyzing the causes of errors.

- Through hierarchical pruning, specifically by removing verification steps at lower layers—we reduce CLoT's resource consumption by 41.8%.

- Based on CLoT, an instruction dataset named CLoT-Instruct is constructed to enable models to learn the "Backward Verification" approach.

- We conduct extensive experiments to validate the effectiveness of CLoT. We compare it with baseline methods across six different tasks, and CLoT performs better in most of them.

## 2 RELATED WORK

Since training LLMs requires enormous resources, effectively improving model performance without retraining has become a key research focus. Wei et al. (2022) introduced the concept of CoT reasoning, emphasizing the importance of deriving conclusive answers through multi-step logical pathways. Kojima et al. (2022) found that simply adding the phrase "let's think step by step" to prompts enables LLMs to perform zero-shot logical reasoning without any additional human-designed prompts. Subsequently, Wang et al. (2023b) proposed Self-Consistency (SC) to replace the greedy decoding strategy. Zhang et al. (2023) developed an automatic CoT framework based on the input problem, eliminating the instability caused by manual prompting. Fu et al. (2023) employed complexity-based multi-step reasoning estimation to execute CoT. Zhu et al. (2025) further enhanced performance by applying pre-prompting in the form of a plugin. However, standard chain-like reasoning still struggles with highly complex logical problems. Variants such as Tree-of-Thoughts Yao et al. (2023), Graph-of-Thoughts Besta et al. (2024), and Markov-chain-inspired rea-

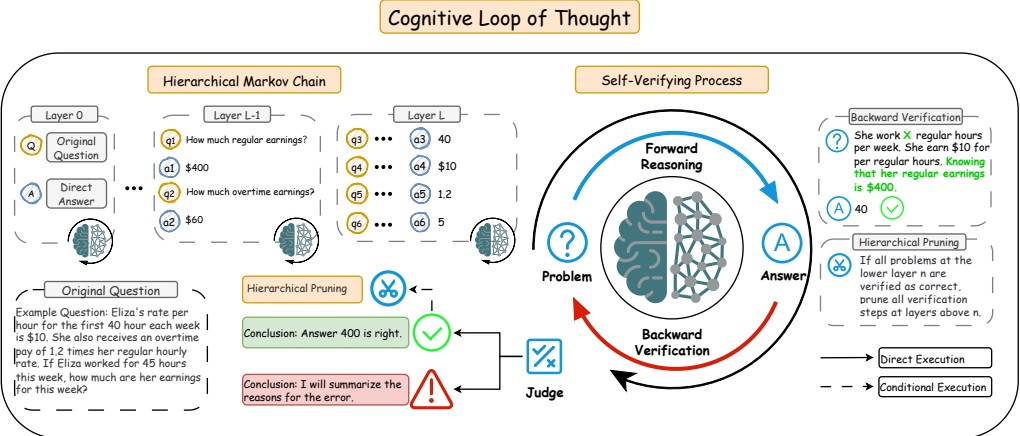

Figure 1: The overview of CLoT. The left side illustrates our hierarchical Markov process, where the top level consists of the original problem and the answer obtained via direct CoT reasoning. Subsequent layers perform step-by-step verification. The right side depicts the backward verification process: the model's generated answer is treated as a known condition, while one of the conditions from the original problem statement is treated as unknown and inferred through reasoning. Finally, reasoning is performed on this modified problem to determine whether the answer derived from the second round of reasoning is equivalent to the originally unknown condition in the problem statement. This design enables CLoT not only to detect incorrect answers but also to pinpoint and correct errors, thereby significantly enhancing the model's safety and reliability.

soning frameworks have effectively enhanced the model's reasoning capabilities Teng et al. (2025). Nevertheless, in practical applications, large models often fail to recognize their own errors and tend to repeat mistakes, limiting their applicability in real-world scenarios.

To expand the application scenarios of LLMs by enhancing their ability to identify and correct errors, researchers have developed various methods to improve their self-correction capabilities. Wang et al. (2023a) verifies answers by integrating inputs from both humans and other models. Weng et al. employs backward verification to evaluate and score multiple candidate answers generated during a tree-of-thought reasoning process. Li et al. (2024) enables models to effectively distinguish between confident and uncertain responses using a simple "Self-Verifying" prompt (e.g., "If you are very confident about your answer, maintain your answer. Otherwise, update your answer"). Zheng et al. (2024) relies on ground truth to determine if a question needs to be answered again. Chen & Li (2024) identifies and corrects errors by having the LLMs analyze the intermediate steps of a CoT for mistakes and pinpoint the source of the error. While these methods have improved self-correction to varying degrees, they often face the challenges summarized by Yang et al. (2024): (1) reliance on high-quality external feedback, (2) lack of comparison with baselines that consume the same number of tokens, and (3) prompts that are strictly designed and potentially complex.

To address these challenges, we propose the CLoT (Loop of Thought) method. Our approach achieves self-correction without requiring any external high-quality feedback. It demonstrates a significant performance improvement compared to a baseline with equivalent token consumption and operates in a zero-shot manner, eliminating the need for designing complex prompt examples.

## 3 METHOD

In this section, we provide a detailed description of the design of the CLoT method, as shown in Figure 1. CLoT guides the model to perform backward verification, identifies potential failures in backward verification, and ultimately summarizes the causes of errors and corrects them. Additionally, we construct a CLoT-Instruct Dataset based on the CLoT approach to help train models to learn "Backward Verification" approach.

## 3.1 Reversible Hierarchical Markov Chain

For a mathematical problem $q$, we observe that successful reasoning involves not only a sequential simplification of the problem through derivation steps but also an implicit hierarchical organization of cognitive processes—such as conceptual abstraction, subproblem decomposition, and strategic planning. To capture this structure, we propose a Hierarchical Markov Chain framework, which models reasoning as a multi-level stochastic process where each level corresponds to a distinct granularity of problem understanding or solution strategy. Let the original problem be denoted by $q_1^{(L)}$ at the max layer $l = L$. At each lower layer $l < L$ (down to $l = 1$), the problem is refined into increasingly concrete and actionable representation $q_t^{(l)}$, fforming a hierarchical structure that transitions from abstract strategies to specific derivations. State transitions within each layer satisfy the Markov property, while cross-layer transitions are jointly governed by top-down refinement and bottom-up generalization mechanisms. Specifically, given a state $q_{t-1}^{(l)}$ and a derivation step $s_{t-1}^{(l)}$ at layer $l$, the next state is generated via:

$$p(q_t^{(l)} \mid q_{t-1}^{(l)}, s_{t-1}^{(l)}), \tag{1}$$

and inter-layer propagation is modeled as:

$$p(q_t^{(l+1)} \mid q_t^{(l)}) \quad \text{(abstraction)}, \qquad p(q_t^{(l)} \mid q_t^{(l+1)}) \quad \text{(refinement)}, \tag{2}$$

this hierarchical structure allows the model to maintain long-range dependencies across reasoning steps through high-level semantic anchors, while still enabling local decision-making at fine-grained levels. Crucially, unlike standard Markov chains, we introduce reversibility in the reasoning trajectory by incorporating a backward verification process, ensuring logical consistency and reducing error accumulation. Given a generated forward reasoning path:

$$\tilde{q}_1 \to \tilde{q}_2 \to \cdots \to \tilde{q}_T \to a = \tilde{q}_{T+1}, \tag{3}$$

where each state $\tilde{q}_t$ represents the problem representation at step $t$ (the sub-problem and the sub-answer), and is associated with a corresponding derivation step $s_{t-1}$, our goal is to conduct a systematic credibility assessment of this single reasoning trajectory. With the above assumption, we have the following objective:

$$\mathcal{L}_{\text{RHMC}} = \sum_{t=1}^{T} \log p(s_t, \tilde{q}_{t+1} \mid \tilde{q}_t) + \sum_{t=1}^{T} \log p^{\leftarrow}(s_t, \tilde{q}_t \mid \tilde{q}_{t+1}), \tag{4}$$

this score consists of two components:

- **Forward term** $\log p(s_t, \tilde{q}_{t+1} \mid \tilde{q}_t)$: measures the plausibility of deriving the next state $\tilde{q}_{t+1}$ from the current state $\tilde{q}_t$, reflecting local deductive validity.
- **Backward term** $\log p^{\leftarrow}(s_t, \tilde{q}_t \mid \tilde{q}_{t+1})$: evaluates whether the preceding state $\tilde{q}_t$ can be reasonably reconstructed via backward inference given the result $\tilde{q}_{t+1}$—i.e., "Is this step logically justifiable?"

Notably, the backward model is not a mere inversion of the forward step, but a semantic-level logical backtracking mechanism. It requires the model to assess whether the predecessor state $\tilde{q}_t$ must exist and remain consistent if the consequent $\tilde{q}_{t+1}$ holds true. This enables the detection of invalid assumptions, non sequiturs, or unjustified leaps in reasoning. By integrating both directional coherence and hierarchical modeling, CLoT provides a fine-grained diagnostic signal that identifies weak or inconsistent reasoning steps within a single chain, enabling self-consistency checking without requiring retraining or external supervision.

## 3.2 Hierarchical Pruning

Building upon the Reversible Hierarchical Markov Chain (RHMC) framework, we introduce a dynamic inference-time verification strategy that exploits the hierarchical dependency structure of reasoning to reduce redundant backward validation.

In hierarchical reasoning, logical correctness propagates downward: if a high-level abstraction (e.g., problem decomposition or strategic planning) is fully consistent under bidirectional validation, then

all lower-level derivations must be logically compatible, assuming sound refinement and abstraction functions. This insight enables us to safely skip detailed verification at fine-grained layers when top-level coherence is confirmed, significantly accelerating the auditing process without sacrificing reliability.

Based on the Equation 4, considering a reasoning hierarchy with $L$ layers, where layer $L$ denotes the most abstract level and layer 1 the most concrete. Let $\mathcal{V}^{(l)}$ denote the backward validation target at layer $l$, defined as:

$$\mathcal{V}^{(l)} = \sum_{t=1}^{T_l} \log p^{\leftarrow}(s_t^{(l)}, \tilde{q}_t^{(l)} \mid \tilde{q}_{t+1}^{(l)}), \tag{5}$$

where $T_l$ is the number of reasoning steps in layer $l$. This score quantifies how well each step in layer $l$ can be justified in reverse, serving as a local measure of logical defensibility.

Crucially, due to the upward propagation of errors, failure at any low-level layer will almost certainly manifest as inconsistency at higher levels. Therefore, if the top layer achieves sufficient backward consistency—i.e., $\mathcal{V}^{(L)} \geq \tau$, where $\tau$ is a threshold calibrated on a validation set. We infer that the entire reasoning chain is globally coherent, and lower layers need not be individually verified.

Our pruning algorithm operates during post-generation verification as follows:

1. **Abstraction:** Generate the full reasoning trajectory at the highest layer $L$: $\tilde{q}_1^{(L)} \to \cdots \to \tilde{q}_{T_L}^{(L)}$.

2. **Validation:** Compute the backward score $\mathcal{V}^{(L)}$ by applying reverse reasoning along the abstracted path.

3. **Pruning:** If $\mathcal{V}^{(L)} \geq \tau$, conclude that the full reasoning hierarchy is logically sound; skip verification of all lower layers.

4. **Refinement:** If $\mathcal{V}^{(L)} < \tau$, descend to layer $L - 1$ and repeat the validation process recursively until either consistency is confirmed or the base layer is reached.

This approach transforms the verification workflow into an adaptive, early-exit protocol. Instead of exhaustively evaluating every step in every layer, we prioritize global coherence checks and only drill down when necessary. Therefore, the overall objective function of CLoT is as follows:

$$\mathcal{L}_{\text{CLoT}} = \sum_{t=1}^{T} \log p(s_t, \tilde{q}_{t+1} \mid \tilde{q}_t) + \mathbb{I}(\mathcal{V}^{(L)} < \tau) \cdot \sum_{l=1}^{L-1} \mathcal{V}^{(l)}, \tag{6}$$

where $\mathbb{I}(\cdot)$ is the indicator function. Notably, the backward terms for layers $l < L$ are only computed when top-level validation fails.

In practice, this reduces the average computational cost of full-chain auditing from $\mathcal{O}(L \cdot T)$ to $\mathcal{O}(T)$ in cases where high-level reasoning is correct—making deep, multi-step verification scalable even for complex problems.

### 3.3 CLoT-Instruct Dataset

#### 3.3.1 Dataset Construction

Building upon the theoretical framework of the RHMC, we introduce CLoT-Instruct, a specialized instruction-tuning dataset designed to train language models with bidirectional reasoning and self-verification capabilities. Unlike conventional cot datasets that focus solely on forward reasoning, CLoT-Instruct explicitly encodes the backward verification mechanism central to our CLoT framework, enabling models to develop intrinsic self-consistency checking abilities.

Theoretically, our construction methodology is applicable to any mathematical reasoning dataset, as it does not rely on any specific problem type but instead builds upon a general hierarchical reasoning structure and bidirectional verification logic. Currently, we have implemented and released three high-quality subsets based on this approach, using GSM8K, SVAMP, and AddSub as the foundational datasets. The construction of CLoT-Instruct follows a two-stage pipeline designed to jointly ensure the quality of forward reasoning and the integrity of backward verification:

- Forward Reasoning: Creation of hierarchical problem-solving trajectories
- Backward Verification: Generation of logically inverted validation questions

### 3.3.2 DATASET STRUCTURE

Formally, let $\mathcal{D}_{\text{origin}}$ denote the original mathematical reasoning dataset, where each instance follows the format $\tau_1 = (q_1, s_{1:T}, a)$. Here, $q_1$ represents the initial problem, $s_{1:T}$ denotes a sequence of $T$ reasoning steps ($T \geq 1$), and $a$ is the final answer.

In contrast to conventional approaches that treat the entire reasoning chain as a monolithic unit, our CLoT framework explicitly decomposes reasoning into hierarchical components, each augmented with bidirectional verification signals. To support this, we construct the CLoT-Instruct dataset, in which every sample is uniformly structured as:

$$\mathcal{X} = \Big( q_{\text{origin}}, \ a_{\text{gt}}, \ \{\tau^{(l)}\}_{l=0}^L, \ max\_layer \Big), \tag{7}$$

where $q_{\text{origin}}$ is the original problem, and $a_{\text{gt}}$ is its ground-truth answer. $max\_layer$ denotes the maximum reasoning depth required. For each layer $l \in \{0, 1, \ldots, L\}$, $\tau^{(l)}$ represents the reasoning trajectory at that layer, defined as

$$\tau^{(l)} = \Big( q^{(l)}, \ s_{1:T_l}^{(l)}, \ \big\{ (q_{\text{verify},t}^{(l)}, a_{\text{verify},t}^{(l)}) \big\}_{t=1}^{T_l} \Big), \tag{8}$$

where $q^{(l)}$ is the problem at layer $l$ (either the original question or a sub-question), $s_{1:T_l}^{(l)}$ is its sequence of reasoning steps, and each pair $(q_{\text{verify},t}^{(l)}, a_{\text{verify},t}^{(l)})$ constitutes a bidirectional verification instance: $q_{\text{verify},t}^{(l)}$ is a logically inverted question derived from the forward step, and $a_{\text{verify},t}^{(l)}$ is its expected answer. This unified representation enables hierarchical dependency tracking, dynamic verification, and scalable supervision across reasoning depths—while maintaining a consistent and extensible data schema.

## 4 EXPERIMENT

In this section, we conduct comprehensive experiments to thoroughly investigate CLoT through extensive benchmark evaluations on six standard datasets, efficiency analysis, data result analysis, and ablation studies.

### 4.1 EXPERIMENTAL SETTINGS

#### 4.1.1 DATASETS

We evaluate CLoT using gpt-4o-mini and gpt-4-1106-preview as backbone models. Our evaluation covers three types of reasoning tasks: mathematical QA reasoning (MATH$_{900}$, Hendrycks et al. (2021), GSM8K$_{1319}$, Cobbe et al. (2021), Addsub$_{395}$, Hosseini et al. (2014), and SVAMP$_{1000}$, Patel et al. (2021)), mathematical multiple-choice (AQUA$_{254}$, Ling et al. (2017)), and commonsense reasoning (CommonsenseQA$_{1220}$, Talmor et al. (2019)). For the large datasets Math and SVAMP, we adopt the same data processing procedure as in Study Chen & Li (2024), evaluating the first 900 and 1,000 questions from their respective test sets. On mathematical datasets, CLoT employs exact numeric matching—i.e., an answer is considered correct only if it exactly matches the numeric value in the original question. On commonsense datasets, it uses synonym-based judgment—i.e., an answer is deemed correct if it is a synonym or near-synonym of the answer in the original question.

#### 4.1.2 BASELINE

Our baselines include classical prompting methods (CoT, Kojima et al. (2022)), CoT with Self-Consistency (CoT-SC, n = 5, Wang et al. (2023b)), CoT with Iterative Summarization Pre-Prompting (ISP-CoT, Zhu et al. (2025)), Analogical Reasoning (AR, Yasunaga et al. (2024)), Complex-CoT (C-CoT, Fu et al. (2023)) and Thought Rollback (Tr, Chen & Li (2024)).

Table 1: Accuracy of CLoT and baselines on six mainstream datasets. Higher is better for all metrics. The best result is highlighted in **bold**, and the second best is underlined.

| Models | AddSub | GSM8K | SVAMP | MATH | AQuA | CSQA | Avg. |
|---|---|---|---|---|---|---|---|
| gpt-4o-mini | | | | | | | |
| CoT | 94.9 | 90.9 | 93.4 | 78.3 | 81.4 | 81.5 | 86.7 |
| CoT-SC(n=5) | 96.1 | 92.0 | 93.5 | **81.0** | 83.1 | **83.4** | 88.3 |
| AR | 82.3 | 91.7 | 92.4 | 73.1 | 77.5 | 78.7 | 81.0 |
| C-CoT | 97.2 | 92.4 | 93.4 | 76.9 | 83.3 | 81.0 | 87.4 |
| ISP-CoT | 96.1 | 92.4 | 93.9 | 77.9 | 83.7 | 81.9 | 87.7 |
| CLoT(ours) | **99.0** | **94.6** | **94.9** | 80.7 | **85.8** | 82.3 | **89.6** |

Table 2: Accuracy of CLoT and baselines on gpt-4-1106-preview

| Models | AddSub | GSM8K | SVAMP | MATH | AQuA | CSQA | Avg. |
|---|---|---|---|---|---|---|---|
| gpt-4 | | | | | | | |
| CoT | 98.0 | 93.1 | 93.4 | 73.0 | 82.7 | 84.9 | 87.5 |
| CoT-SC(n=5) | 98.4 | 94.2 | 93.5 | **81.6** | 83.1 | **85.6** | 89.4 |
| AR | 87.1 | 93.4 | 92.7 | 73.1 | 79.5 | 79.1 | 84.2 |
| C-CoT | 98.4 | 94.2 | 93.6 | 80.6 | 82.7 | 83.9 | 88.9 |
| ISP-CoT | 97.7 | 93.3 | 93.5 | 77.7 | 84.2 | 83.4 | 88.3 |
| Tr | 97.5 | 94.2 | 91.3 | 71.9 | 79.9 | 79.4 | 85.7 |
| CLoT(ours) | **99.0** | **95.4** | **95.0** | **81.6** | **86.9** | 84.9 | **90.5** |

## 4.2 MAIN RESULTS

Main results are presented in Table 1 and Table 2. CLoT consistently achieves state-of-the-art performance across all datasets and both model backbones (gpt-4o-mini and gpt-4-1106-preview). Under the gpt-4o-mini setting, CLoT attains the highest average accuracy of 89.6%, outperforming the strongest baseline CoT-SC by 1.3%. Notably, CLoT sets new records on AddSub (99.0%), GSM8K (94.6%), SVAMP (94.9%), and AQuA (85.8%), demonstrating its robustness across both arithmetic and complex multi-step reasoning tasks.

When scaled to the more capable gpt-4, CLoT further widens its lead, achieving the highest average accuracy among all methods. It matches or exceeds the best baseline on every individual dataset, with particularly significant gains on GSM8K (+1.2%), SVAMP (+1.5%), and AQuA (+3.8%) compared to the previous best results. These improvements underscore CLoT's ability to effectively leverage iterative generate-and-verify cycles to refine reasoning paths, especially in domains where initial solutions are prone to subtle errors (e.g., AQuA and MATH).

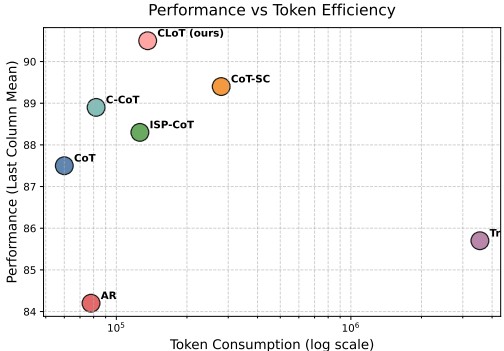

Figure 2: The results of CLoT efficiency. The token consumption corresponds to the total number of tokens used to solve the same 100 gsm8k problems (idx0 – idx99).

Moreover, CLoT maintains strong performance even on challenging symbolic and mathematical reasoning tasks, matching the best baseline (CoT-SC) while significantly outperforming other iterative or corrective approaches. The consistent gains across diverse problem types, ranging from simple arithmetic AddSub to multi-hop commonsense CSQA and math, highlight the generality and adaptability of our verification-driven framework.

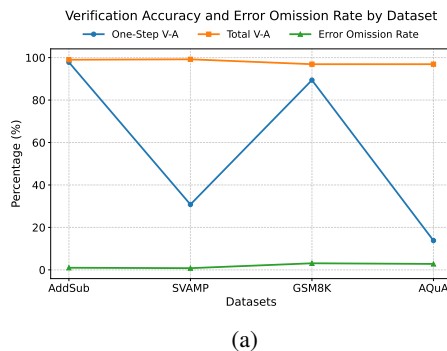 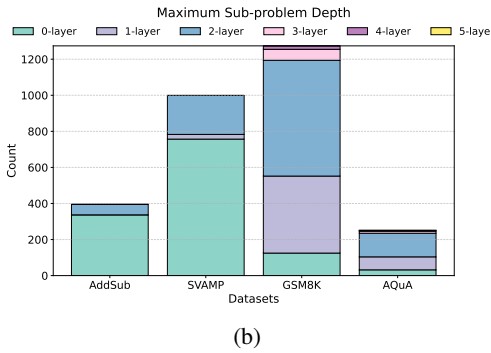

(a)             (b)

Figure 3: We present a visualization of the dataset analysis. Figure (a) shows the various accuracy metrics on the dataset, where V-A denotes Verification Accuracy. Figure (b) shows the maximum number of layers in the multi-layer Markov chains used in the dataset.

### 4.3 EFFICIENCY ANALYSIS

We further evaluate the efficiency of different methods in terms of token consumption. The results are shown in Figure 2 and were obtained using the gpt-4o-mini API. Traditional methods such as CCoT and CoT-SC consume approximately 280k tokens. In contrast, Tr uses chain-of-thought for self-verification and requires as many as 3.3M tokens, resulting in very high computational costs. AR is more token-efficient, using only 78k tokens. However, its overall performance still lags behind other approaches. Notably, CLoT achieves performance on par with or even better than existing methods while consuming just 136k tokens. This highlights its strong reasoning efficiency under low token overhead. This efficiency stems from its hierarchical pruning strategy, which selectively removes redundant verification steps at lower reasoning layers. This design enables high reasoning efficacy with minimal token overhead, making CLoT particularly suitable for large-scale and resource-constrained applications.

### 4.4 DATASETS ANALYSIS

To systematically evaluate our model's verification capability on mathematical reasoning, we built verification datasets from four standard math QA benchmarks: AddSub, SVAMP, GSM8K, and AQuA (see Figure 3). We analyze four key metrics:

- One-Step Verification Accuracy: the fraction of problems whose initial solution is both correct and verified on the first attempt.

- Total Verification Accuracy: the proportion of problems that are either correctly solved or whose errors are successfully detected.

- Error Omission Rate: the fraction of incorrect solutions missed by the verifier.

One-Step Verification Accuracy is high on AddSub (97.8%) and GSM8K (89.4%), indicating strong self-verification on problems with simple or clear reasoning paths. It drops sharply on SVAMP (30.8%) and AQuA (13.8%), reflecting the difficulty of generating verifiable solutions in a single pass for semantically complex or multi-step problems—often requiring iterative refinement.

In contrast, Total Verification Accuracy remains consistently high across all datasets, demonstrating that CLoT reliably ensures output correctness by either producing correct answers or flagging errors. Correspondingly, the Error Omission Rate is low, confirming the verifier's high recall. Notably, AQuA achieves an omission rate of only 2.8%, comparable to GSM8K, despite its low one-step accuracy. This highlights the effectiveness of multi-round verification in catching errors even from poor initial solutions. Finally, the maximum layer counts are shown in Figure 3b. The results indicate that AddSub and SVAMP have at most two layers of subproblems, whereas GSM8K and AQuA exhibit three or more layers, demonstrating that more challenging tasks require multiple rounds of the "generate-and-verify" cycle.

Table 3: Ablation study results of CLoT based on gpt-4o-mini. The experiment used four datasets and token consumption to evaluate the contributions of three components. The token consumption corresponds to the total number of tokens used to solve the same 100 GSM8K problems (idx0 – idx99).

| Models | Token | AddSub | GSM8K | SVAMP | AQuA | Avg. |
|---|---|---|---|---|---|---|
| CoT (baseline) | 60k | 94.9 | 90.9 | 93.4 | 81.4 | 90.2 |
| CoT + SV | 98k | 92.0 | 93.5 | 87.3 | 81.4 | 88.6 |
| CoT + HMC | 112k | 94.9 | 90.9 | 93.4 | 81.0 | 90.1 |
| CoT + RHMC | 325k | 94.4 | 93.5 | 93.9 | 84.6 | 91.6 |
| CoT + RHMC + HP | 136k | **99.0** | **94.6** | **94.9** | **85.8** | **93.6** |

## 4.5 ABLATION ANALYSIS

To systematically evaluate the effectiveness of each component in our proposed method, we conduct a series of ablation studies using the gpt-4o-mini model. As shown in Table 3, we incrementally introduce four key designs: (1) standard Chain-of-Thought (CoT) with a single direct answer verification (CoT + Self-Verifying, or CoT + SV), serving as the baseline; (2) CoT enhanced with Hierarchical Markov Chain decomposition (CoT + HMC) without any verification, to assess the benefit of structured reasoning alone; (3) CoT with Reversible Hierarchical Markov Chain (CoT + RHMC), which adds step-by-step verification at each reasoning level to improve overall reliability; and (4) CoT + RHMC further augmented with Hierarchical Pruning (HP), where lower-level verification steps are skipped once all subproblems at a higher level are confirmed correct.

The results show that CoT + SV achieves limited performance across datasets, with an average accuracy of 88.6%. Adding the HMC structure yields slightly unstable results but produces more organized reasoning paths. With RHMC, the model shows clear improvements on AddSub, GSM8K, and SVAMP, reaching an average accuracy of 91.6%. Finally, our full method (CoT + RHMC + HP) further boosts average accuracy to 93.6% while reducing total token consumption from 325k to 136k (41.8%). This demonstrates that hierarchical pruning effectively balances efficiency and performance. Overall, the results confirm the importance of layered verification and dynamic pruning in complex reasoning tasks.

## 5 CONCLUSION

We introduce Cognitive Loop of Thought (CLoT), a novel reversible hierarchical reasoning framework that mimics human cognition: after forward decomposition into hierarchically dependent sub-problems, CLoT performs backward verification at each level to validate intermediate steps. Crucially, once higher-level sub-problems are confirmed correct, CLoT prunes redundant lower-level computations—reducing inference cost by 41.8% while boosting accuracy. We further curate CLoT-Instruct, the first instruction dataset explicitly teaching backward verification. Evaluated across four mathematical reasoning benchmarks, CLoT achieves 99.0% accuracy on AddSub with GPT-4o-mini, outperforming standard CoT and CoT-SC by 4.1% and 2.9%, respectively. Our work demonstrates that integrating cognitive-inspired reversibility and hierarchical pruning not only mitigates error accumulation but also enables efficient, trustworthy, and interpretable reasoning in large language models.

## ETHICS STATEMENT

This work is a theoretical and empirical study focused on improving the reasoning efficiency of large language models. It does not involve human subjects, personally identifiable information, or sensitive data. All experiments were conducted using publicly available datasets and standard APIs, in compliance with their terms of service. The proposed method, CLoT, aims to reduce computational costs and improve scalability, which may contribute to more sustainable and accessible AI systems. We do not anticipate any direct harmful societal impacts from this research. However, as with any advancement in language model capabilities, we acknowledge the potential for misuse and encourage responsible deployment in real-world applications.

## REPRODUCIBILITY STATEMENT

We strive to ensure the reproducibility of our work. All code, model checkpoints, and detailed experimental configurations used in this study are publicly available at: `https://anonymous.4open.science/r/CLoT-7EBD`. The datasets we used are either publicly accessible (with citations provided in Section 4.1.1). We also include a requirements.txt file specifying all dependencies, and provide scripts to reproduce the main results reported in the paper. We welcome feedback and encourage independent verification of our findings.

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

## A LLM USEGE

Large Language Models (LLMs) were employed exclusively as a linguistic aid during the preparation of this manuscript. Their role was limited to supporting the authors in refining the phrasing, enhancing textual coherence, and improving the overall readability of the paper. Specifically, the LLM assisted with sentence restructuring, grammatical correction, and smoothing transitions between ideas—thereby strengthening the clarity and fluency of the exposition.

## B ANSWER FORMATION

Here is the CLoT process dialogue, with different steps handled through CoT. Table 6 and Table 3 show the reasoning processes for mathematical problems. In the final part of our reasoning to generate the answer, we will fix the format of the final answer, as shown in Table 4.

| Task | Answer-format Instructions |
|---|---|
| GSM8K, SVAMP, AddSub, math | You can freely reason in your response, but please enclose the final answer within <answer></answer>tags(pure number without units and explanations) |
| AQuA, CommonsenseQA | You can freely reason in your response, but please enclose the final option within <answer></answer>tags(pure uppercase option without explanations) |

Table 4: Answer-format instructions for different tasks.

| SVAMP Problem |
|---|
| Q: Kylar went to the store to buy glasses for his new apartment. One glass costs $5, but every second glass costs only 60% of the price. Kylar wants to buy 16 glasses. How much does he need to pay for them? |
| **Step 1**, First CoT |
| Q: Lets think step by step. |
| ***Thought*** |
| <answer> 64 </answer> |
| **Step 2**, Backward Verification |
| Q: Please replace one numerical value in the original problem with X and incorporate the answer as a known condition. |
| Kylar went to the store to buy glasses for his new apartment. One glass costs X, but every second glass costs only 60% of the price. Kylar wants to buy 16 glasses. Knowing that he need to pay $64 for them. What's the X? |
| **Step 3**, Solve the new problem |
| Q: Lets think step by step. |
| ***Thought*** |
| <answer> 5 </answer> |
| **Step 4**, Judge |
| 5 = 5 |
| **Answer**: 64 |

Table 5: Examples of Successful Verification by CLoT on Mathematical Problem Datasets.

| **SVAMP Problem** |
| --- |
| Q: Kylar went to the store to buy glasses for his new apartment. One glass costs \$5, but every second glass costs only 60% of the price. Kylar wants to buy 16 glasses. How much does he need to pay for them? |
| **Step 1**, First CoT |
| Q: Lets think step by step.

***Thought***

<answer> 96 </answer> |
| **Step 2**, Backward Verification |
| Q: Please replace one numerical value in the original problem with X and incorporate the answer as a known condition.

Kylar went to the store to buy glasses for his new apartment. One glass costs X, but every second glass costs only 60% of the price. Kylar wants to buy 16 glasses. Knowing that he need to pay \$64 for them. What's the X? |
| **Step 3**, Solve the new problem |
| Q: Lets think step by step.

***Thought***

<answer> 10 </answer> |
| **Step 4**, Judge |
| $4 \neq 5$ |
| **Step 5**, Decompose sub-problems |
| Q: You are teasked with breaking down a math problem reasoning process into sub-problems.

q1,q2,q3,... |
| **Step 6**, For each subproblem, iteratively execute Steps 1–4. |
| If all subproblems at the same level pass verification, exit the loop. |
| **Step 7**, Second CoT |
| Q: Lets think step by step. Please refer tp the following sub-problems and sub-answers.

***Thought***

<answer> 64 </answer> |
| **Answer**: 64 |

Table 6: Examples of CLoT's verification failures on mathematical problem datasets.

---

**Algorithm 1:** CLoT Reasoning

---

**Input:** Original problem $q_1^{(L)}$ at top layer $L$
**Output:** Reasoning trajectory $\{\tilde{q}_1, \ldots, \tilde{q}_{T+1}\}$ and score $\mathcal{L}_{\text{RHMC}}$

**Initialization:** Set $t \leftarrow 1$, current layer $l \leftarrow L$, and initial state $q_1^{(L)}$;
**while** $t \leq T$ **do**
    // Intra-layer transition (derivation)
    Sample next state $q_t^{(l)} \sim p(q_t^{(l)} \mid q_{t-1}^{(l)}, s_{t-1}^{(l)})$;
    // Cross-layer propagation
    **for** $l$ *from L down to* 1 **do**
        Refine: $q_t^{(l)} \sim p(q_t^{(l)} \mid q_t^{(l+1)})$;
        Abstract: $q_t^{(l+1)} \sim p(q_t^{(l+1)} \mid q_t^{(l)})$;
    **end**
    // Record forward trajectory
    Set $\tilde{q}_t \leftarrow q_t^{(l)}$ with derivation step $s_t$;
    $t \leftarrow t + 1$;
**end**

**Forward reasoning path:** $\tilde{q}_1 \to \tilde{q}_2 \to \cdots \to \tilde{q}_T \to a = \tilde{q}_{T+1}$;

**Scoring with reversibility:**;
$\mathcal{L}_{\text{RHMC}} \leftarrow 0$;
**for** $t = 1$ **to** $T$ **do**
    // Forward plausibility
    $\mathcal{L}_{\text{RHMC}} \leftarrow \mathcal{L}_{\text{RHMC}} + \log p(s_t, \tilde{q}_{t+1} \mid \tilde{q}_t)$;
    // Backward justification
    $\mathcal{L}_{\text{RHMC}} \leftarrow \mathcal{L}_{\text{RHMC}} + \log p^{\leftarrow}(s_t, \tilde{q}_t \mid \tilde{q}_{t+1})$;
**end**
**return** $\{\tilde{q}_1, \ldots, \tilde{q}_{T+1}\}, \mathcal{L}_{RHMC}$

---

## C  ALGORITHM

To better understand our approach, we summarize CLoT's reasoning process in the following algorithm 1:

