# OpenReview forum: "Cognitive Loop: Reversible Hierarchical Markov Chain for Bidirectional Self-Verifying Reasoning"
_ICLR.cc/2026/Conference — ICLR 2026 Conference Withdrawn Submission_

### Official Review · Reviewer_18ot · 2025-10-29

**Soundness:** 2
**Presentation:** 3
**Contribution:** 1
**Rating:** 2
**Confidence:** 4

**Summary:**

The authors introduce Cognitive Loop of Thought (CLoT), where the authors build on chain of thought reasoning work using hierarchical markov chain through the decomposition of a problem into sub-problems. At every step, a backward verification mechanism is used to avoid errors throughout the thinking process, mimicking human thinking.

**Strengths:**

- Proposed framework outperforms several COT baselines proposed in the literature.
- The framework is evaluated on 6 different reasoning benchmarks, covering three types of reasoning tasks, with consistent results across most of them.
- Efficiency analysis shows that CLoT consumes less tokens than several other baselines, while still achieving high performance.

**Weaknesses:**

- Has this been tested on reasoning-tuned LLMs? It seems that the experimental set-up only looks at GPT-4. Although results are good, this does not necessarily mean that this works on other LLMs.
- It seems that CLoT is most effective on mathematical reasoning. Why is this case? More discussion needs to be included on this point.

**Questions:**

See above.

---

> ### Author Response · Authors · 2025-11-20
> **Thank you for your careful review—it has greatly contributed to improving the quality of the paper.**
>
> **Q1:** We are currently supplementing results from other large models, including DeepSeek and Gemini. The specific results will be added in the revised version.
>
> **Q2:** Our method primarily focuses on mathematical reasoning, as self-verification can only effectively leverage original numerical values as hard metrics for validation within the mathematical domain. In conventional QA tasks, it is difficult to confirm the correctness of the original reasoning chain using self-verification, leading to insignificant performance gains. We are also actively exploring more general self-verification methods.

---

> > ### Comment · Reviewer_18ot · 2025-11-22
> >
> > Thank you for the clarifications and for outlining the additional experiments you plan to include. Extending the evaluation to other large models will certainly strengthen the paper, and your explanation regarding the method’s focus on mathematical reasoning is appreciated. My evaluation, however, is based on the originally submitted manuscript, and I will not be changing my score.

---

### Official Review · Reviewer_93F3 · 2025-10-30

**Soundness:** 3
**Presentation:** 3
**Contribution:** 3
**Rating:** 6
**Confidence:** 4

**Summary:**

The paper introduces a new reasoning framework for LLMs.
It is designed to improve the accuracy and reliability of multi-step reasoning by mimicking the human cognitive process of verifying one's own work.
They propose to verify both forward and backward and design a hierarchical pruning to reduce token cost.
The authors validate CLoT on six mathematical and commonsense reasoning benchmarks, demonstrating better performance.

**Strengths:**

1. The core concept mimics human reasoning behavior and the verification is a novel method for self-correction.
2. The paper considers the trade-off between efficiency and effectiveness and CLoT creates a balance between them.
3. The empirical results prove the effectiveness of the concept.

**Weaknesses:**

1. The backward verification process is easy for mathematical problems, but for more complex tasks, the backward question could be vague and hard to define.
2. Although the authors propose a instruct dataset, the effectiveness of training on such dataset is not reported.

**Questions:**

1. How does the CLoT-Instruct dataset contribute to the training of the LLM?
2. The reversible hierarchical Markov chain relies on decomposing the problem into sub-problems with hierarchical dependencies. How are these hierarchies and sub-problems initially generated? Is this decomposition an automatic process performed by the LLM, and if so, how sensitive is the final accuracy of CLoT to the quality of this initial decomposition step?
3. How does the process differ from deductive reasoning paradigm [1][2], apart from the backward verification?
4. Typos:
    - Line 171: fforming

[1] Ling, Zhan, et al. "Deductive verification of chain-of-thought reasoning." Advances in Neural Information Processing Systems 36 (2023): 36407-36433.

[2] Zhu, Tinghui, et al. "Deductive beam search: Decoding deducible rationale for chain-of-thought reasoning." arXiv preprint arXiv:2401.17686 (2024).

---

> ### Author Response · Authors · 2025-11-20
> **Thank you for your careful review—it has greatly contributed to improving the quality of the paper.**
>
> **Q1:** CLoT-Instruct can provide self-verification data for the mathematical domain, which is currently lacking. This helps models learn self-verification and enhances their credibility. In the revised version, we will present models fine-tuned on our CLoT-Instruct dataset to demonstrate its effectiveness.
>
> **Q2:** During the first step of model decomposition into subproblems, we prompt the model to output dependency relationships among subproblems. These relationships are then used to establish a hierarchy—the greater the number of dependencies, the higher the importance of the problem. We have not yet studied how the quality of decomposition affects final accuracy. In the revised version, we will add ablation experiments using models of different tiers for decomposition to evaluate their impact on final results.
>
> **Q3:** The main differences between our approach and existing subproblem decomposition methods lie in two aspects: reverse verification and the hierarchical Markov chain construction method. Specifically, we prompt the model to output dependency relationships among subproblems and use the quantity of these dependencies to determine hierarchical levels.

---

### Official Review · Reviewer_r8AS · 2025-11-01

**Soundness:** 3
**Presentation:** 3
**Contribution:** 3
**Rating:** 2
**Confidence:** 4

**Summary:**

The paper proposes a novel reasoning framework Cognitive Loop of Thought (CLoT) for large language models (LLMs) that mimics human cognitive verification. Unlike conventional Chain-of-Thought (CoT) methods that rely on forward-only reasoning, CLoT introduces a Reversible Hierarchical Markov Chain (RHMC) that alternates between forward reasoning and backward verification. Each problem is decomposed into hierarchical sub-problems; the model validates each reasoning step by reversing the logic—treating conclusions as known and re-deriving premises. A pruning strategy further skips verification of lower layers once higher-level consistency is confirmed, cutting inference cost by 41.8%. The authors also construct CLoT-Instruct, a dataset that teaches backward verification. Experiments on six benchmarks (AddSub, GSM8K, SVAMP, MATH, AQuA, CommonsenseQA) show consistent accuracy gains over CoT, CoT-SC, and other baselines, achieving 99.0% on AddSub with GPT-4o-mini and 90.5% average accuracy on GPT-4

**Strengths:**

1. The paper introduces a cognitively inspired, reversible reasoning paradigm, and provides rigorous mathematical formulation and efficient hierarchical pruning.
2. It demonstrates strong empirical improvements with reduced token usage.

**Weaknesses:**

1. The validation is only limited to reasoning benchmarks, and the experiment section lacks generalization tests on non-mathematical or other real-world tasks.
2. Does the backward verification assumes deterministic reversibility? Hierarchical Pruning assumes that if high-level reasoning passes backward verification, then all lower-level steps are also correct. Does upper-level reasoning happen to appear logically consistent while masking subtle arithmetic or semantic errors in lower layers?
3. The pruning decision depends on a calibrated threshold $\tau$ for backward consistency. If $\tau$ is too low, false positives occur (invalid reasoning passes verification); if too high, pruning rarely triggers, negating efficiency gains. Any robust and adaptive method for tuning $\tau$?
4. The overall novelty is incremental given some previous work, e.g., Atom of thoughts for Markov llm test-time scaling, and the work is missing some important references, e.g., Markov chain of thought for efficient mathematical reasoning.

**Questions:**

Refer to weaknesses.

---

> ### Author Response · Authors · 2025-11-20
> **Thank you for your careful review—it has greatly contributed to improving the quality of the paper.**
>
> **Q1:** Our method primarily focuses on mathematical reasoning because self-verification techniques can only leverage original values as hard metrics for matching validation within the mathematical domain. In conventional QA domains, self-verification struggles to confirm the correctness of the original reasoning chain, hence yielding insignificant results. We are also actively exploring general-purpose self-verification methods.
>
> **Q2:** We would like to clarify that **in the context of mathematical reasoning, backward verification is uniquely robust against such “hidden error” scenarios**, precisely because of the deterministic and reversible nature of mathematical operations.
>
> When we perform backward verification (e.g., substituting the final answer back into the original problem statement or intermediate expressions), we are not merely checking logical consistency — we are verifying *numerical equivalence* against ground-truth values derived from the problem itself. For example:
>
> > If the problem asks: “If $ x + 3 = 7 $, what is $ x $?”, and the model outputs $ x = 4 $, backward verification computes $ 4 + 3 = 7 $ — which matches the original equation exactly.
> > Any arithmetic error (e.g., $ x = 5 $) would immediately fail this check.
>
> This means that **a correct high-level conclusion cannot be achieved without correct low-level computations** — because mathematical correctness is compositional and cumulative. A single misstep in arithmetic, algebraic manipulation, or unit conversion will propagate and invalidate the backward verification result.
>
> Thus, while your concern is theoretically sound in domains with ambiguous semantics or probabilistic logic (e.g., commonsense QA or natural language inference), it is **less applicable in mathematical reasoning**, where backward verification acts as a *hard constraint* rather than a soft consistency check.
>
> **Q3:** We apologize for any misunderstanding caused by our previous description. In fact, the true pruning condition is strict matching: we validate every step within the top-level reasoning chain. Only when every step perfectly matches the ground truth do we conclude that the hierarchy is reliable and skip verification of lower layers. This maximizes reduction of false negatives.
>
> **Q4:** Unlike traditional Markov chains, CLoT prompts the model during the initial subproblem decomposition step to output dependency relationships among subproblems, then layers them according to dependency count—the higher the count, the greater the importance. This forms a hierarchical Markov chain enabling hierarchical pruning. CLoT delivers dual improvements in efficiency and credibility within prompting domains. Additionally, we contribute an instruction dataset to help models learn how to perform backward verification.

---

### Official Review · Reviewer_zxbC · 2025-11-03

**Soundness:** 3
**Presentation:** 2
**Contribution:** 3
**Rating:** 6
**Confidence:** 3

**Summary:**

The paper proposes CLoT (Cognitive Loop of Thought), which is a reasoning method where the model generates forward chain-of-thought and also checks if each step can be logically reversed. If a step cannot be reversed to recover the previous information, the model identifies it as incorrect and revises only that part instead of restarting everything (a form of self-correction). The reasoning is organized from high-level plans to detailed steps, so verification begins at the top and only goes deeper if needed, making the process efficient. This approach improves accuracy and uses fewer tokens compared to standard CoT or self-consistency methods.

**Strengths:**

1. The work is well inspired and well motivated
2. It is good to see that the authors have not only chased after final performance, but rather showed that this improvement comes with same number of tokens
3. The dataset released would be useful to the community
4. The idea, from what I can tell is a novel contribution

I do have some critiques, Please see weaknesses

**Weaknesses:**

1. The approach seems math specific ( or at least specific to deductive reasoning). While this is true, the title and abstract does not mention this explicitly. And this is a concern for me.
2. I can understand the use of LLMs to polish a paper for grammar and typos (which the authors also declared in appendix), but I feel In the process of polishing with LLMs, this has gotten unnecessarily verbose.
3. Further It is unclear to me why so much mathematical equations are used in the main body of the paper, while the concept can probably be explained in much simpler words ? For example I see equation 4 defined a   L_rhmc, but I cant see any algorithm directly optimizing it, or even using it as metric.
4. It is hard to understand the method in detail unless I see the prompts, so I would request the authors to kindly share them with the reviewers, and later add to the main paper.

**Questions:**

Please see weaknesses.

---

> ### Author Response · Authors · 2025-11-20
> **Thank you very much for your kind and thoughtful evaluation of our paper.**
>
> Your feedback is invaluable in helping us improve the quality of our work.
>
> **Q1:** Indeed, CLoT is primarily designed for mathematical scenarios. We are actively exploring more robust general-purpose methods, which will be presented in future versions of this work.
>
> **Q2 & Q3:** In the revised version, we will refine the language to reduce unnecessary formulas and make the paper more accessible to a broader audience. For example, we will simplify or rephrase Equations 4 and 6 to enhance clarity without sacrificing technical rigor.
>
> **Q4:** We have included a basic prompt template in the appendix. We are happy to incorporate a more complete and detailed prompt template into the main paper. The core prompt used to construct the hierarchical Markov chain is as follows:
>
> > You are tasked with breaking down a math problem reasoning process into sub-questions. Please ensure that there are no repeated questions.
> >
> > **Instructions:**
> > 1. Break down the reasoning process into a series of sub-questions.
> > 2. Each sub-question should:
> >    - Be written in interrogative form.
> >    - Have a clear numerical answer.
> >    - List the indexes (0-based) of other sub-questions it depends on (can be an empty list).
> > 3. Dependencies are defined as information needed to answer the current sub-question that:
> >    - Does NOT come directly from the original question.
> >    - MUST come from the answers of previous sub-questions.

---

### Note · Authors · 2026-01-02

**Comment:**

Thank you to every reviewer. We will improve our research work according to the review comments. Looking forward to seeing you again and wishing everyone academic success.

**Withdrawal Confirmation:**

I have read and agree with the venue's withdrawal policy on behalf of myself and my co-authors.